# Surface, Structural, and Mechanical Properties Enhancement of Cr_2_O_3_ and SiO_2_ Co-Deposited Coatings with W or Be

**DOI:** 10.3390/nano12162870

**Published:** 2022-08-20

**Authors:** Mihail Lungu, Daniel Cristea, Flaviu Baiasu, Cornel Staicu, Alexandru Marin, Oana Gloria Pompilian, Bogdan Butoi, Claudiu Locovei, Corneliu Porosnicu

**Affiliations:** 1National Institute for Laser, Plasma and Radiation Physics, 077125 Magurele, Romania; 2Department of Materials Science, Faculty of Materials Science and Engineering, Transilvania University, 500068 Brasov, Romania; 3Faculty of Physics, University of Bucharest, 077125 Magurele, Romania; 4Ken and Mary Alice Lindquist Department of Nuclear Engineering, Penn State University, University Park, State College, PA 16802, USA; 5Surface Analysis Laboratory, Institute for Nuclear Research Pitesti, 115400 Mioveni, Romania; 6Magnetism and Superconductivity Laboratory, National Institute of Materials Physics (NIMP), 077125 Magurele, Romania

**Keywords:** metal-oxide, thin films, morphology and roughness, chemical state, crystalline structure, hardness and adhesion, chromium oxide, silicon dioxide, tungsten, beryllium

## Abstract

Direct current (DC) and radio frequency (RF) magnetron sputtering methods were selected for conducting the deposition of structural materials, namely ceramic and metallic co-depositions. A total of six configurations were deposited: single thin layers of oxides (Cr_2_O_3_, SiO_2_) and co-deposition configurations (50:50 wt.%) as structural materials (W, Be)—(Cr_2_O_3_, SiO_2_), all deposited on 304L stainless steel (SS). A comprehensive evaluation such as surface topology, thermal desorption outgassing, and structural/chemical state was performed. Moreover, mechanical characterization evaluating properties such as adherence, nano indentation hardness, indentation modulus, and deformation relative to yielding, was performed. Experimental results show that, contrary to SiO_2_ matrix, the composite layers of Cr_2_O_3_ with Be and W exhibit surface smoothing with mitigation of artifacts, thus presenting a uniform and compact state with the best microstructure. These results are relevant in order to develop future dense coatings to be used in the fusion domain.

## 1. Introduction

In the fusion context, the final design of barrier coatings and plasma-facing components are far from being established, as future fusion power stations (e.g., DEMOnstration power plant) impose extreme unprecedented conditions [1]. The R & D of novel combinations of alloys relevant to the fusion domain that could withstand numerous destructive phenomena determined by high flux plasma, radiation, low permeability, and neutron loads is still mandatory to be addressed [1]. Despite their significance, a constant delay was observed in proposing and studying novel permeation barriers designated for hydrogen embrittlement reduction [2,3]. The design of structural barrier coatings applied as part of a complex tritium recovery system is primarily oriented on mitigating the amount of tritium permeation yield by integrating the proper dense and defect-free materials and coatings [4].

In this paper, we focused our attention on hydrogen permeation barrier-like candidates (HPB), namely oxides and metals, proposing different configurations and investigating their structural integrity in order to provide a general insight prior to complex permeation measurements campaigns. Generally, oxides express different response to hydrogen permeability [5], thus they are highly applicable as permeation barrier candidates and as structural barrier materials in hydrogen storage devices [6], vacuum solar receivers [7], and fusion reactors [8,9]. Taking these facts into consideration, two oxides with different properties in relation to the hydrogen permeation were studied, namely chromium oxide with an applicable permeation reduction factor up to ~10^3^ [2,10,11,12] and the less studied silicon dioxide [13]. The coatings were obtained in pure and co-deposition configurations with well-known plasma fusion relevant materials, namely tungsten [14,15,16,17] and beryllium [18,19,20].

The proposed mixed-material layers could provide some structural integrity insight that could be linked to the plasma post-exposure conditions where co-deposition phenomena are studied in the plasma–wall interaction among other material migration mechanisms (e.g., erosion, transport, and deposition) [21]. To the best of our knowledge, there are no reports in the scientific literature concerning these co-deposited configurations (Cr_2_O_3_ and SiO_2_, each co-deposited with Be or W). Nevertheless, some studies at the layer-substrate interface were reported, namely mechanical stress concerns related to the metallic oxides’ coatings deposited on SS substrate, that occurs due to the differences of thermal expansion coefficient, which can further cause defects [22] translated into cracks and localized delamination reducing the permeation mitigating yield.

Radio frequency (RF) magnetron sputtering was used to deposit the samples presented herein, due to its demonstrated applicability in the deposition of oxides layers of Cr_2_O_3_ [23] and SiO_2_ [24,25,26,27,28] with good permeation properties. Direct current (DC) magnetron sputtering was used for releasing the W and Be atoms from their respective targets. The coatings were deposited on an austenitic 304L stainless steel substrate, often being used in permeation measurements [29]. Moreover, the experimental plan was based on the previously established validation protocol which correlates the configuration of the layers to the mechanical properties (i.e., adhesion, hardness), surface morphology (i.e., roughness), chemical state, and crystalline structure, all applied to (W, Be)—(Al_2_O_3_, Er_2_O_3_) configurations [30]. Hereinafter, the same protocol is applied for (W, Be)—(Cr_2_O_3_, SiO_2_) configurations, expanding the scientific insight regarding the structural integrity while contributing to the development of dense and compact coatings.

## 2. Experimental Details

### 2.1. Sample Preparation

Depositions were carried out in a conventional magnetron sputtering facility in co-deposition configuration integrating magnetrons powered up with RF power sources for oxides targets and DC power sources for the metallic targets. Details regarding the experimental configuration were disseminated elsewhere [30].

The sputtering gas (Ar) was introduced at a flow rate between 15–20 sccm, while the distance between substrate and magnetron target was kept at ~10 cm. Specific variable parameters such as source power and the applied current for each proposed deposition configuration are described in Table 1. The oxide targets (ø 2-inch, 3 mm thickness) of high purity Cr_2_O_3_ (99.99%) and SiO_2_ (99.90%) (Neyco vacuum and materials, Neyco, Vanves, France) and attached to Cu backing plates to increase the heat transfer, avoid target cracking, and maintain a stable RF plasma discharge. Metallic targets with the same geometrical configuration were used for the co-deposition process: Be (99.95%, Goodfellow Cambridge Ltd., Huntingdon, UK) and W (99.99%, MaTeck Material-Technologie, and Kristalle GmbH, Julich, Germany).

The coatings were deposited onto ø 40 mm 304L stainless steel (SS) disks, and 10 × 10 mm^2^ silicon and carbon wafers (Goodfellow Cambridge Ltd., Huntingdon, UK), all types polished to a mirror-like finish. The 304L stainless steel (SS) disks were chosen in order to fulfill the requirements of tribological evaluation such as indentation, evolution of the friction coefficient (CoF) as function of distance, and adhesion/cohesion behavior. The silicon and carbon substrates provide better and smoother coatings for roughness measurements, morphology, and structural analyses.

In situ quartz microbalance monitor (QCM, Inficon, Bad Ragaz, Switzerland) and scanning electron microscopy (SEM) cross-sectional images were acquired for sample thickness validation. In order to achieve the initially proposed 50:50 wt.% concentration in the co-deposition configurations, the sputtering rate was limited by the maximum applicable anode current, a necessary limitation to avoid thermal stress-based failure of the oxides targets. The imposed limitation does not affect the integrity of the coatings, usually higher deposition rates being avoided due to the increased occurrence of structural artefacts (e.g., cracks, delamination, and voids) [31]. Moreover, here we mention that the lower value of the pressure condition found for the SiO_2_ system was mandatory in order to achieve a higher deposition rate to obtain a final sample thickness in the micrometer range.

### 2.2. Analytical Techniques

Atomic force microscopy (AFM) analysis in contact mode was performed on the TT-2 model apparatus (Quantum Design S.A.R.L., Les Ulis, France), overviewing defect free and smooth selected regions of 5 × 5 µm^2^. Image post-processing for root mean square roughness calculations were performed with the Gwyddion v 2.6 software (Department of Nanometrology, Czech Metrology Institute, Czech Republic [32]).

Scanning electron microscopy (SEM) measurements in cross-sectional view were performed in order to observe the morphology variations and also for validating the QCM results on the deposition rate. SEM and energy dispersive X-ray (EDX) analyses were performed on an FEI Inspect S instrument (Thermo Fisher Scientific, Hillsboro, OR, USA), at a working voltage up to 30 kV and a working distance up to 30 mm.

X-ray photoelectron spectroscopy (XPS) investigations were performed on an Escalab 250 system (Thermo Scientific, East Grinstead, UK) equipped with a monochromated AlKα (1486.6 eV) X-ray source and a base pressure in the analysis chamber of 10^−8^ Pa. The energy scale was referenced to the Au4f7/2 line at a binding energy of 84.0 eV. The acquired spectra were calibrated with respect to the C1s line of surface adventitious carbon at 284.8 eV. An electron flood gun was used to compensate for the charging effect in insulating samples.

X-ray diffraction (XRD) analysis was performed using a Bruker D8 Advance diffractometer (Bruker Corporation, Karlshruhe, Germany) powered at 40 mA and 40 kV (λCuK_α_ = 0.154 nm). The XRD results were acquired in symmetric geometry between 2θ = 15–65° at room temperature and an angular step of 0.02°. All measurements were carried out on carbon-based substrates, thus avoiding the use of Si.

Quantification for outgassing process of atomic and molecular species was conducted by means of thermal desorption spectroscopy (TDS) measurements. These measurements were performed at a maximum temperature of 1000 °C during a predefined period (<2 h). The applied quadrupole mass spectrometer (QMS) for analyzing the desorbed elements consisted in Pfeiffer Vacuum QME 220 (Pfeiffer Vacuum GmbH, Asslar, Germany), while de mass measuring range was reduced to 1–300 a.m.u in order to fine-tune the scanning acquisition rate, accordingly.

Instrumented indentation measurements were performed using a NHT^2^ device (CSM Instruments/Anton Paar, Peseux, Switzerland), with a Berkovich diamond tip (tip radius = 100 nm), with at least 20 imprints per sample, up to the maximal load of 5 mN and a 10 mN/min loading/unloading rate, while the post-processing of resulted data was based on the dedicated method developed by Oliver, W.C., et al. [33].

The adhesion/cohesion behavior was analyzed with a micro-scratch instrument (CSM Instruments/Anton Paar, Peseux, Switzerland) equipped with a Rockwell diamond stylus with a 100 µm tip radius. At least six scratch tracks were obtained on each sample, with a 4 mm length. The applied load was progressively increased from 0.03 up to 15 N, with a loading rate of 7.5 N/min and a speed of 2 mm/min. Thresholds on critical loads were imposed for describing the appearance of the first cracks, the first delamination (partial removal of coating), and more than 50% delamination, which was considered as total coating failure.

The evolution of the friction coefficient (CoF) as function of distance was investigated using a ball-on-disk tribometer in rotation mode (CSM Instruments/Anton Paar, Peseux, Switzerland). Experiments were conducted at room temperature using Si_3_N_4_ balls as the friction couple, a 1 N applied load, a sliding speed of 1 cm/s, and the stop condition as 3 m of sliding. Prior to the friction tests, both friction couples (samples and Si_3_N_4_ balls) were ultrasonically cleaned in isopropanol and blow dried, to remove any traces of impurities.

## 3. Results

### 3.1. Surface Topology Characterization

The films’ surface features were investigated through SEM and AFM observations. The AFM height images and the root-mean-square (RMS) roughness showing the topology changes between pure oxides and metal-reinforced configurations are presented in Figure 1. The results for the A1 (Cr_2_O_3_) sample are not shown herein, due to the high surface roughness which is over the capability of the instrument and that would contribute to high displacement on the Z direction of the cantilever integrated in the measuring setup. Distinctive topology variations were observed for film configurations including Be (i.e., A3 and B3) while the other studied configurations have a surface roughness below 6 nm, that could be an indication of the coatings following the irregularities of the Si substrate during the deposition process (RMS Si = 4 nm), as reported elsewhere [34]. As reference, here we mention that profilometry measurements were carried on the SS substrate prior deposition with the determined values: Ra = 0.18 μm, Rz = 1.09 μm and Rq= 0.22 μm.

In comparison with the pure chromium oxide (sample B1), the presence of Be in the co-deposition configuration (sample B3) seems to lead to an increase in surface roughness, while the presence of W (samples A2 and B2) significantly reduced the RMS over the investigated surface, down to the detection limit. The trend for W to form surfaces with dense and compact particularities is in line with existing reports on relatively similar co-deposition configurations [35]. Considering the SiO_2_ configuration (series B), smooth surfaces deposited by RF magnetron sputtering were expected [36]. In contrast, the presence of Be in the B3 sample determined a slight increase in RMS value by a notable amount (~50%), while a slight decrease was observed for the B2 configuration. Generally, it seems that the presence of W tends to homogenize the co-deposited samples at the surface level, while Be seems to promote a relatively higher surface roughness, as one could observe in Figure 1.

The surface morphology acquired by SEM, depicts the typical microstructure of coatings deposited by magnetron sputtering without any gold surface precoating. One can observe in Figure 2 the morphology differences such as deposition defects expressed as thin flakes, detached layers, or grains with random orientation (A1); randomized nucleation with conical growth (A2) and grains (A3); smooth surface without any particularities related to shape or size (B1); formation of isolated grains (B2); high roughness with visible defects illustrated as cavities (B3)

Furthermore, the porosity fraction for the B3 configuration was determined in the range of 2–3% by image analysis on the investigated surface (559 µm^2^). This particularity of grain distribution and grains agglomeration could be partially explained due to insufficient atom diffusion distance based on the low deposition rate and residual stress specific to Be during DC magnetron sputtering deposition [37]. Moreover, the residual stress could be further influenced by the thermal expansion difference between the oxide and Be [38]. To some extent, this effect could be mitigated by means of increasing the sputtering rate of Be, which is not of interest in this work, in order to maintain the initially proposed elemental ratio configuration.

The chemical composition was analyzed qualitatively using EDX, taking into consideration the limitations regarding the detection of Be. Measurements indicate that the sputtered oxides (Cr_2_O_3_, SiO_2_) contain O (wt.%) at close values to the atomic ratio in stoichiometric configuration, confirming the coexistence of metal-oxide in a bulk state (Table 2), excepting the B3 configuration. The detected traces of Fe are associated with SS substrate. Moreover, no elemental conglomeration could be observed during EDX mapping measurements, confirming the homogeneity of the deposited coatings.

### 3.2. Chemical State, Structure, and Thermal Desorption Measurements

In order to study the chemical interaction of atoms during the deposition process, high resolution XPS spectra of the main elements present at the surface were acquired.

Figure 3a shows the overall chemical behavior of silicon as a function of Be and W addition indicating a chemical shift towards higher binding energy side, from 98.9 to 103.5 eV, respectively. This illustrates that there is a gradual oxidation process of Si on the surface in the presence of Be and W atoms. Moreover, by peak-fitting the Si2p envelope (Figure 3c,e,g), it was possible to determine the oxidation states of Si in the samples. Thus, the peaks located at 98.9 eV [39,40] and 103.5 eV [39,40] are attributed to unoxidized Si and fully oxidized SiO_2_, while intermediate states are found between the former and the latter. The two suboxides of Si are peaked at 102.5 eV [39,40] and 103.0 eV [39,40] which can be associated to a stoichiometry x between 1.2–1.6 and 1.5–1.9, respectively.

Therefore, the silicon chemistry indicated a non-oxidized form of silicon only in the presence of beryllium (Figure 3g) whilst Si sub-oxidation states can be observed in the presence of both Be and W (Figure 3e,g). The reason for this partial oxidation of silicon is probably related to insufficient oxygen in the system since both beryllium and tungsten reacted with oxygen (Figure 4a,c).

Interestingly and from the quantitative point of view, the oxygen content decreases from ~75% to ~58% only when beryllium was introduced into the sample matrix (Table 3).

Like the previous discussion, the overview picture in Figure 3b reveals a similar chemical behavior of chromium, showing a mixture of its metallic and oxidized states, the former occurring only in the presence of beryllium. After curve-fitting the Cr2p_3/2_ peak, the extracted binding energies of 574.3 eV [39], 575.5 eV [39], 576.7 eV [39], 577.5 eV [39] and 578.9 eV [39] are corresponding to metallic Cr, CrO_2_, Cr_2_O_3_, Cr(OH)_3_, and CrO_3_, respectively.

Furthermore, incorporating Be and W during the deposition process induces the occurrence of a lower oxidation state of chromium, Cr^4+^ (CrO_2_), in addition to Cr^6+^ (CrO_3_). As a consequence, beryllium acts as an oxygen-getter by halving the amount of chromium oxide features and on the other hand, protects and promotes the metallic state of chromium (Figure 3h). This is in excellent agreement with silicon chemistry where beryllium reduces the silicon oxide contribution, favoring its unoxidized form (Figure 3g).

At the same time, tungsten has a similar chemical behavior in both systems regardless of the environment, suggesting an approximately two-times increase in its metallic feature, as compared to beryllium (Figure 4). Thus, the W4f doublet was decomposed into four components, located at 31.3 eV [39,41], 32.3 eV [39,41], 34.2 eV [39,42], and 36.0 eV [39,42] characteristic to W^0^, W^4+^, W^5+^, and W^6+^, respectively (Figure 4c,d).

Quantitatively, the oxygen drops to the minimum value of 58% after the incorporation of Be into the samples (Table 3). Moreover, the two “dopants”, Be and W, are found in similar concentrations for both Si- and Cr-based samples.

The diffractograms of samples A1, A2, B1, and B2 are presented in Figure 5. The Cr_2_O_3_ and SiO_2_ layers deposited on graphite (C) substrates (ICDD 04-014-0362) show amorphous phase with no specific structural order, as is commonly expected for oxides layers deposited by magnetron sputtering at room temperature [43,44].

For sample A2 the β-W phase (ICDD 00-047-1319) is observed, but the peak corresponding to the (210) diffraction plane has a low intensity with high shift in 2θ position which indicates the formation of a distorted structure. Further, the B2 layer also highlights the presence of β-W phase through the wide shoulder centered around 2θ = 39.85° suggesting a very small crystalline coherence length given by the structural order at low distances.

TDS measurements were used to evaluate the outgassing process of atomic and molecular species trapped on the deposited matrix. In order to conduct the desorption spectra of H_2_O (18), N_2_ (28), O_2_ (32), and CO_2_ (44), configurations deposited on Si substrate (10 × 10 mm^2^) were investigated under a heating rate of 10 °C/min, while overviewing the difference in desorption at peak temperatures specific to the binding forces between elements and trapping regions. For a better understanding of the TDS spectra, it must be specified that, on the left, desorption signal is plotted as a function of time and on the right, the temperature (C) (red line) as a function of time.

Significant desorption of H_2_O (at 400 °C) was observed for A1. In general, desorption of weakly bound peaks associated with CO_2_ could be observed at similar temperatures for all studied configurations. The CO_2_ signal (magenta) is well defined; two peaks can be observed in each graph. The first one is wider between 250 and 500 °C with a maximum release at 400 °C. The second one is thinner, between 50–600 °C. Strong variations of N_2_ in relation to the studied element matrix were observed as follows: constant quasilinear increase reported to temperature in A1; highest desorbed signal at temperature plateau (at 1050 °C) for A2; high content desorption at 400°C was reported for Be co-depositions (i.e., A3 and B3), thus changing the retention mechanism. The raw TDS data suggest that the presence of an oxide bond at the surface has led to higher desorption peak temperatures observable for N_2_ [45], with B3 as exception, results partially correlated with EDX findings.

The presence of N_2_ (blue) in each sample is well noticed by the TDS measurements, the desorption mechanism tends to continue even after the temperature reaches its maximum limit (1050 °C). It is possible to delimit up to four peaks that appear at about the same desorption temperature, only slightly shifted, for A1 (a), A2 (b), A3 (c), B1 (d), and B2 (e). No desorption mechanism differences regarding the O_2_ were observed, with minimal signal detected as observed in Figure 6.

### 3.3. Mechanical Characterization

Considering that the proposed configurations exhibited different chemistry, surface morphology, and structures, some mechanical properties were further investigated, namely: indentation hardness (H), indentation elastic modulus (E), elastic strain to failure ratio (H/E), adhesion critical loads, and friction coefficient. Figure 7 contains the variation of hardness, of the indentation elastic modulus, and the elastic strain to failure ratio (H/E), all as function of the sample configurations, and compared to the uncoated stainless-steel substrate. The performed measurements were conducted with low loads (penetration depth below 10% of the coating thickness), in order to minimize or exclude entirely the influence of the substrate on the results concerning the mechanical behavior relating to H and E.

Some correlations could be extracted from the results from Figure 7: the pure Cr_2_O_3_ oxide is harder than the pure SiO_2_ one, a behavior in agreement with other results [46]; W in co-deposition with oxides tends to significantly influence the mechanical properties of the films by lowering their hardness by a factor of two (samples A2 and B2), while in contrast Be shows a higher probability of increasing to some extent the average hardness; in relation to the variation of the elastic modulus, one can observe that the samples from set B follow the same pattern of the hardness, while the samples from set A exhibit an inverse trend, sample A2 exhibiting the lowest hardness and the highest elastic modulus of the series. The variation of these two parameters (instrumented hardness and elastic modulus) translates to the variation of the elastic strain to failure ratio, a high H/E ratio being often a reliable indicator of good wear resistance in a coating. Based on these predictions, it would seem that the W containing samples would exhibit a lower wear performance, compared to the pure oxides and the ones doped with Be.

The adhesion/cohesion evaluation was conducted by means of micro scratch testing. Generally, the type of failure observed for each studied configuration is dependent on the applied load, substrate properties, and residual stress of the deposited layer. The studied layers had a softer substrate in comparison to the deposited layers, which encourage the formation of buking/spallation failure modes that are useful in order to evaluate the coating to substrate adhesion properties [47]. During the micro scratch tests, a variety of destructive patterns (Figure 8) made by the diamond stylus were observed at the corresponding load values, known as critical loads. Figure 8 contains the micrographs showing the failure of the coatings, while Figure 8e contains the critical loads for these failure events. Here we mention that B1 is considered independently from other configurations based on the thickness differences. Premature heavy delamination of A3 and B2 configurations was observed on the samples, consequently the results for these two samples are not shown herein. One can observe the buckling or wedging spallation for A1, A2, B1, and B3 configurations, respectively. This failure mode corresponds to coatings deposited on a softer substrate (see Figure 7, hardness variation), and the failures could be described from isolated cracks and deformations (e.g., A1 and B1) to hard delamination (e.g., A2 and B3).

The bucking failure mode observed for the A1 sample evolved as a response to the mechanical stress generated by the stylus translation, where the evolved localized elongated cracks perpendicular to the scratch pattern and expose the substrate were directly associated with already present interface defects. Furthermore, the occurrence of wedge spallation failure mode, observed for the A2 and B3 configurations, can be linked to the coating thickness, where the coating becomes too rigid to reduce the stress provoked by the stylus, resulting in hard spallation throughout the layer. Sample B1 expresses a recovery spallation type failure mode, this mode being associated with elastic recovery of the layer and is dependent on the plastic deformation expressed as isolated cracking of the coating on either side of the stylus scratch path.

The variation of the dynamic friction coefficient as function of distance, obtained against Si_3_N_4_ balls, is shown in Figure 9. All samples, either from set A or set B, exhibit a variation of the friction coefficient which could be attributed to the stick-slip phenomenon. Moreover, we consider that the evolution of the coefficient of friction is influenced mainly by the nature of the sample surface, the presence of the third body being also very important. Particularly for A1 and B2 configurations we mention that the initial friction coefficients with Si_3_N_4_ balls sliding against deposited films were lower. However, friction coefficients increased as debris formed and transferred to the wear track. The morphology of the wear track changed during the friction test from a smooth to a rough one due to the formed debris. In regards to the Cr_2_O_3_ based films (Figure 9b), the co-deposition with W, and to a higher degree with Be leads to a significant decrease in the coefficient of friction, compared to the pure oxide. This behavior could be linked to the increased ductility caused by Be and W addition. An agreement between the H/E ratio and the lower friction coefficient can be observed, especially in case of samples A3 and B1. Yin-Yu Chang and co-workers reported similar trends on the graded Ti–C:H films [48] or Cr/Cr-W/W-DLC/DLC multilayer coatings [49]. Inversely, in the case of SiO_2_ based films (Figure 9a), the addition of W and Be leads to the increase in the coefficient of friction, more significant in the case of W addition. However, the coefficient of friction of the pure SiO_2_ oxide suddenly increases after 2.6 m, signifying the delamination of the film. Under certain circumstances, it can be said that the W and Be addition contribute to improved mechanical properties of mixed layers based on Cr and Si oxides, in order to develop future dense and compact coatings.

## 4. Conclusions

Several primary conclusions can be drawn from the present work:Surface topology characterization: SEM images pointed out the surface morphology changes for the Cr_2_O_3_ and SiO_2_ oxides, while grain agglomerations promoting pore formations were observed for SiO_2_: Be configuration; Co-deposition with W could mitigate the RMS factor and homogenize the surface, as determined from the AFM measurements;Elemental distribution expressed as wt.% concentration was measured using EDX for oxide configurations (except SiO_2_: Be), thus evaluating the stoichiometry (O and W content); while TDS data suggest that the presence of an oxide bond at the surface has led to higher desorption peak temperatures observable for N2, excepting SiO_2_: Be. Moreover, defects present in the Cr_2_O_3_ matrix determined a significand desorption of H_2_O at 400 °C;XRD results indicated amorphous phase with no specific structural order for the measured configurations, commonly observable for oxides deposited by magnetron sputtering;Silicon and chromium-based systems showed improved protection against oxidation when tungsten and beryllium were embedded into the sample matrix during the deposition process of the films; also, the metallic feature among oxides of silicon and chromium can be seen only in the presence of beryllium.Mechanical evaluation: at nano-scale, W in co-deposition with oxides tends to lower the hardness, while Be shows a higher probability of increasing the average hardness; Tribology measurements determined that for the SiO_2_ based films, the addition of W and Be leads to the increas in the coefficient of friction, while the enhancement of coating hardness is followed by the drawback of low adherence as observed for SiO_2_: W configuration; adherence evaluation provided several predictable results, since the buckling mode is associated with interface defects, wedge spallation is observable for rigid coatings (e.g., Cr_2_O_3_: W) and recovery spallation was observable for SiO_2_ in agreement with indentation results at micro-scale.

## Figures and Tables

**Figure 1 nanomaterials-12-02870-f001:**
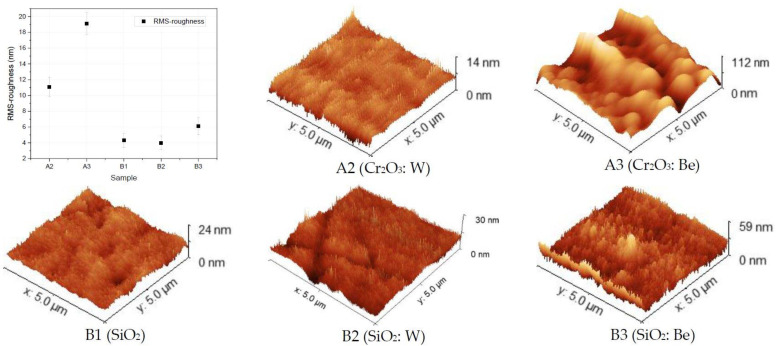
3D AFM images for thin films deposited on Si substrates with associated RMS-roughness (nm) determined values on random selected regions of 5 × 5 µm^2^.

**Figure 2 nanomaterials-12-02870-f002:**
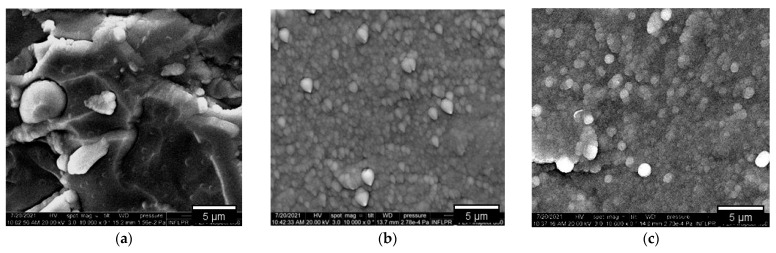
SEM images of oxides and metal-oxides: A1 (**a**)—flake-like structures; A2 (**b**)—randomized cone shape nucleation, <1.5 µm in diameter; A3 (**c**)—sphere/droplet shape nucleation, <2.5 µm in diameter; B1 (**d**) and B2 (**e**)—isolated grain formations, <0.5 µm in diameter; B3 (**f**)—defects expressed as pores isolated by grains, <2.5 µm in diameter.

**Figure 3 nanomaterials-12-02870-f003:**
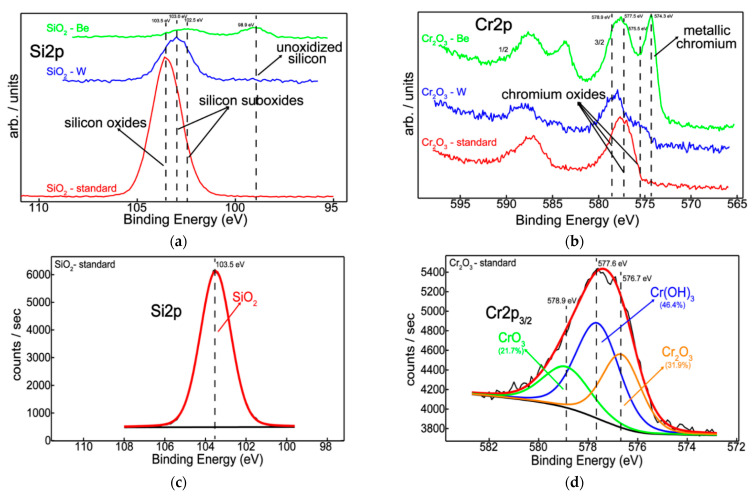
Superimposed XPS spectra for the three stages of Si (**a**) and Cr (**b**) systems; XPS peak-fitted spectra for the three stages of Si (**c**)—B1, (**e**)—B2, (**g**)—B3 and Cr (**d**)—A1, (**f**)—A2, (**h**)—A3 systems.

**Figure 4 nanomaterials-12-02870-f004:**
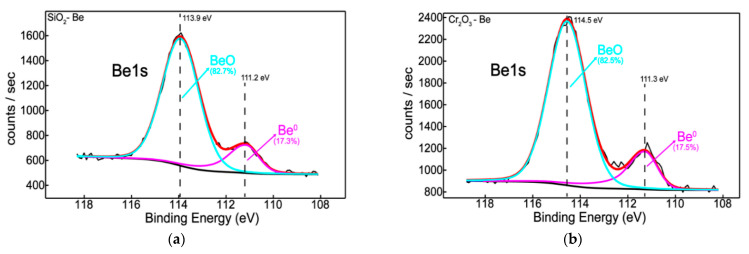
Be1s and W4f XPS peak-fitted spectra for Si- (**a**)—B3, (**c**)—B2 and Cr, (**b**)—A3, (**d**)—A2-based systems.

**Figure 5 nanomaterials-12-02870-f005:**
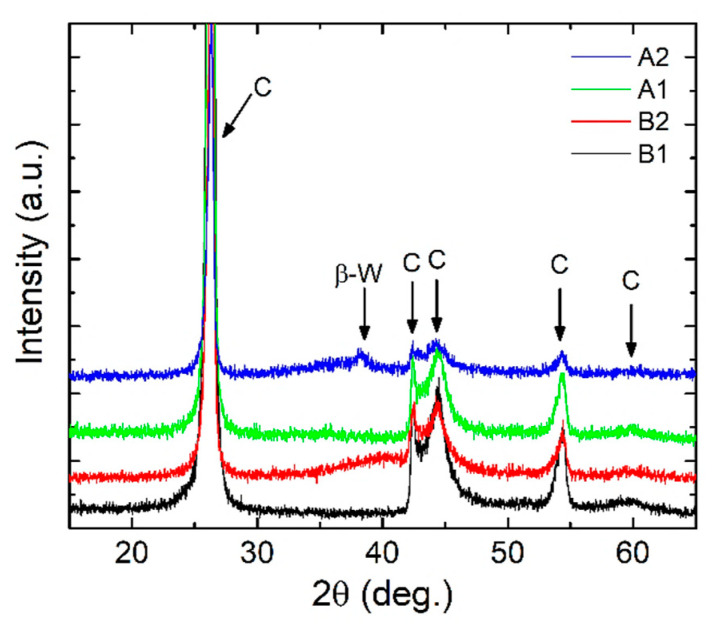
Diffractograms of magnetron sputtered layers: A1, A2, B1, and B2; notation C shows the XRD peaks of the graphite substrates.

**Figure 6 nanomaterials-12-02870-f006:**
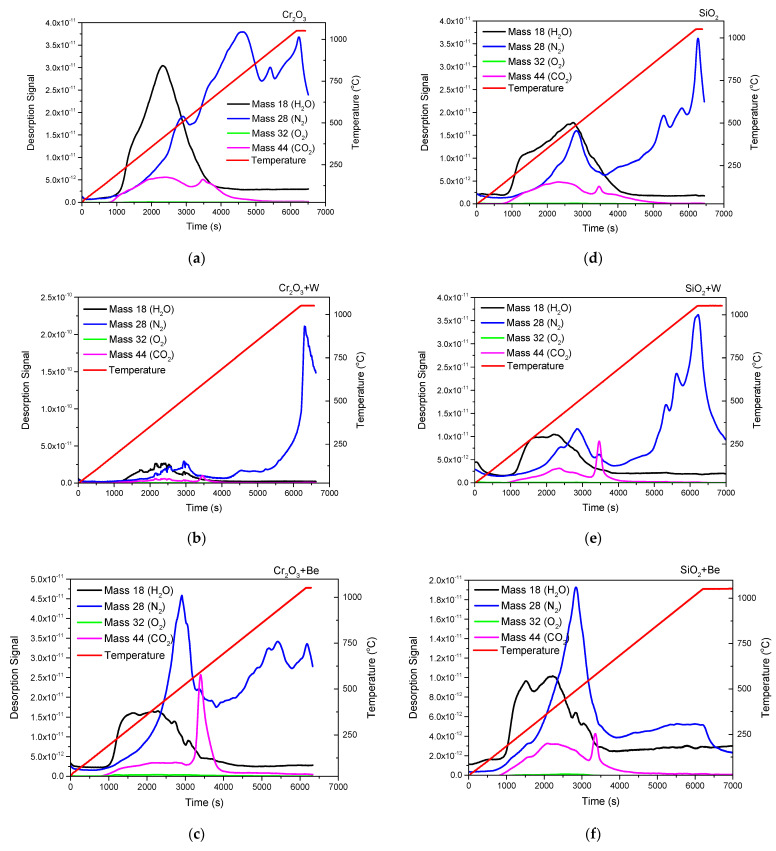
H_2_O, N_2_, O_2_, and CO_2_ desorption profiles as a function of time and at a heating rate of 10 °C/min, acquired for the following samples: A1 (**a**), A2 (**b**), A3 (**c**), B1 (**d**), B2 (**e**), and B3 (**f**).

**Figure 7 nanomaterials-12-02870-f007:**
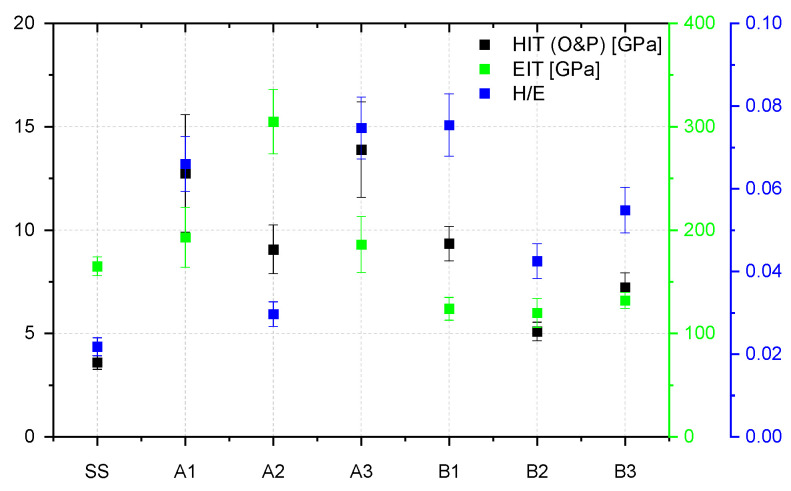
Indentation hardness, young modulus, and elastic strain to failure ratio determined for SiO_2_ and Cr_2_O_3_ investigated configurations deposited on SS substrate analyzed as a reference.

**Figure 8 nanomaterials-12-02870-f008:**
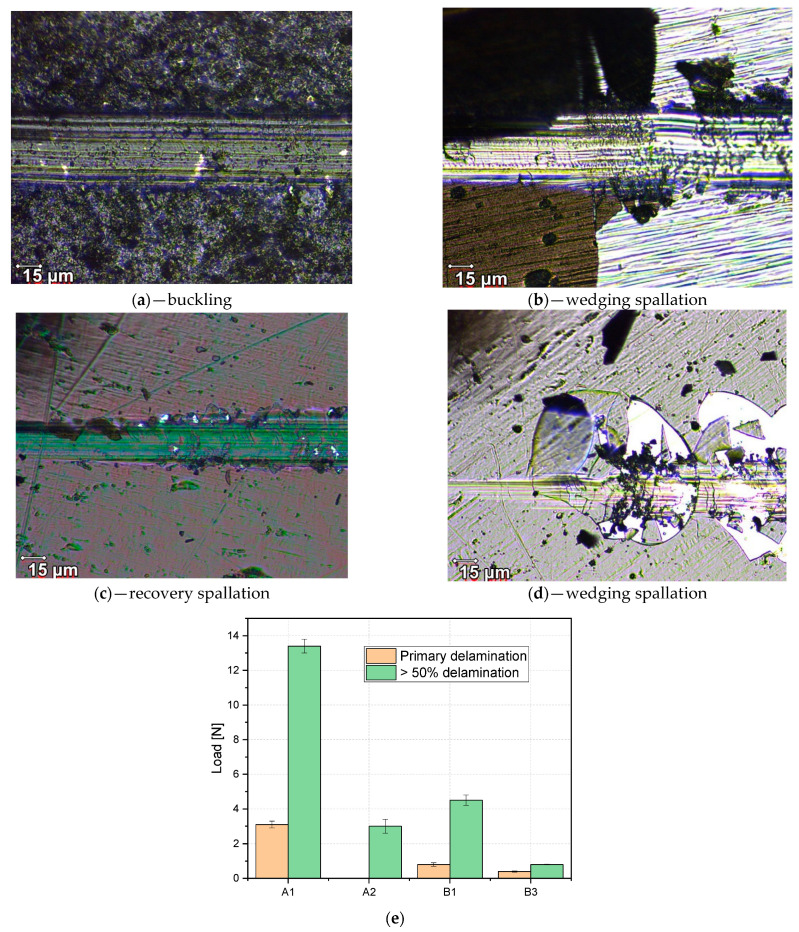
Post-scratch test measurements overview for A1 (**a**), A2 (**b**), B1 (**c**), and B3 (**d**); loads attributed to first delamination and over 50% coating removal (**e**).

**Figure 9 nanomaterials-12-02870-f009:**
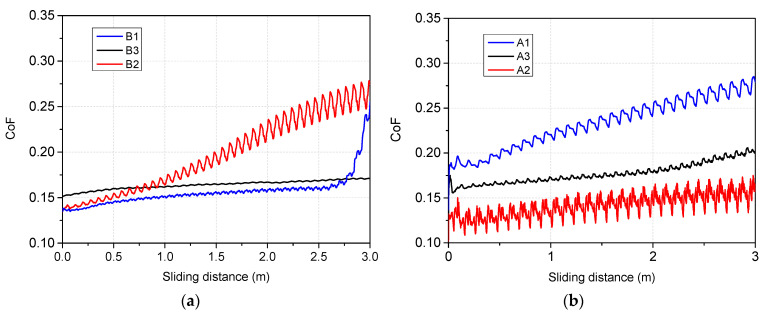
The variation of the friction coefficient as function of wear distance: (**a**) SiO_2_ and (**b**) Cr_2_O_3_ configurations.

**Table 1 nanomaterials-12-02870-t001:** Deposition parameters.

Sample Id	Configuration	RF [W]	DC		Pressure [10^−3^ mbar]	Deposition Rate ×10^−1^ [nm/s]	Validated Layer Thickness [µm]
			U [kV]	I [A]			
**A1**	**Cr_2_O_3_**	100	Not applicable		4	0.2	5
**A2**	**Cr_2_O_3_: W**		0.28	0.04	4	0.2: 0.06	5
**A3**	**Cr_2_O_3_:Be**		0.55	0.15	4	0.2: 0.6	5
**B1**	**SiO_2_**		Not applicable		1.25	0.4	1.6
**B2**	**SiO_2_: W**		0.27	0.02	4	0.4: 0.3	5
**B3**	**SiO_2_: Be**		0.5	0.15	4	0.4: 0.6	5

**Table 2 nanomaterials-12-02870-t002:** EDX composition measurements (wt.%).

Sample	Element	Weight %	Error %
**A1**	O K	31.2	5.7
	Fe L	3.6	42.8
	Cr K	65.1	11.9
**A2**	O K	13.1	21.3
	W M	48.5	10.1
	Cr K	38.4	22.7
**A3**	O K	32.4	5.3
	Fe L	4.6	43.6
	Cr K	63.0	12.7
**B1**	O K	47.6	8.9
	Fe L	3.6	32.3
	Si K	51.3	11.1
**B2**	O K	12.1	16.6
	Fe L	6.4	45.5
	Si K	10.4	16.5
	W M	71.1	10.3
**B3**	O K	47.7	9.1
	Fe L	5.8	26.3
	Si K	46.5	11.6

**Table 3 nanomaterials-12-02870-t003:** XPS determined element relative concentrations (at.%).

Sample	Be1s	O1s	Si2p	W4f	Cr2p
**A1**	-	90.7	-	-	9.3
**A2**	-	75.5	-	18.2	6.3
**A3**	36.2	57.5	-	-	6.3
**B1**	-	74.4	25.6	-	-
**B2**	-	75.8	10.5	13.7	-
**B3**	36.9	58.1	5.0	-	-

## Data Availability

The data presented in this study are available on request from the corresponding author.

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
