# Peer review of "Surface, Structural, and Mechanical Properties Enhancement of Cr2O3 and SiO2 Co-Deposited Coatings with W or Be"

_nanomaterials, 2022, doi:10.3390/nano12162870_

Round 1

Reviewer 1 Report

The manuscript reports experimental investigations on the enhancement of surface, structural, and mechanical properties of Cr2O3 and SiO2 co-deposited coatings with W or Be. Systematic measurements were made on the samples, which add important information to the data base of hydrogen permeation barrier (HPB) coating. However, some concerns raised and need to be discussed by the authors to improve the quality of the paper. The main points are listed below:

 1.      Deposition of SiO2 layer is conducted under a pressure of 1.25·10-3 mbar (Table 1) which is a lower value compared to the other synthesis. How the authors chose this pressure? It would be beneficial if the authors could provide explanation for that.

2.      Some additional experimental details regarding the AFM measurements should be added to the manuscript:

(a)what is the experimental uncertainty in the roughness measurement?

(b)is it possible to demonstrate the reproducibility of such measurements?

3.      The results of chemical composition examined by EDX method (Table 2) are provided with extremely high accuracy. Are these measurements were calibrated with reference materials? Moreover, In page 6, line 199, The authors mentioned “The detected traces of Fe could be residue from the deposition chamber.” What data support this assertion?

Author Response

Dear reviewer,

We would like to thank for your comments and observations. We found these comments as valuable and helpful while improving our paper, and they are also important guidance for our research. Therefore, we have rigorously studied the comments and revised them accordingly, while hoping to receive your acceptance.

Thank you for your consideration,

Sincerely,

The authors,

Reviewer 2 Report

A very good submission by the authors.  Just a spell check/grammar check required prior to resubmission.

Author Response

We would like to thank the referee for the comment and, at the same time, to apologize for the grammar mistakes in the first version of our manuscript. We revised the English form of the manuscript.

Reviewer 3 Report

The paper «Surface, structural, and mechanical properties enhancement of 2 Cr2O3 and SiO2 co-deposited coatings with W or Be» deals with the deposition by DC and RF of Cr- and Si-oxide based coatings, especially for fusion and plasma applications. Different and complementary characterization techniques are used, to provide a comprehensive description of the properties of the investigated materials. Even if the subject is suitable for the publication in the MPDI journal Nanomaterials, major revisions are needed before that the paper is publicly available. The comments are organized in three sections, that is (0) use of English, (1) general comments, and (2) punctual comments.  

0. USE OF ENGLISH. The English is suitable for a correct expression of the concepts, ideas, results and conclusions. No specific comments are made about the use of English and specific expressions.

1. GENERAL COMMENTS. The paper deals with an interesting subject, that is the characterization of specific coatings for plasma applications. The characterization of the substrate should also be presented, because it can affect in many ways the growth and the features of the produced coatings.

As a most important element, the paper is lacking a more robust framework (which are the coatings used in literature for this kind of applications, if any? which are the specifications that such a kind of coating needs to be respected? resistance to hydrogen embrittlement, resistance to tritium permeation, absence of defects,...? thermal resistance? what else?) Which are the features of similar coatings, or coatings with a similar purpose, from literature? which are the ideal property values the authors are aiming at? which are the properties of the coatings presented in literature, when we compare them with the properties obtained in this paper? how far are the authors from the values they want?

«To the best of our knowledge, 57 there are no reports in the scientific literature concerning these co-deposited configura-58 tions (Cr2O3 and SiO2, each co-deposited with Be or W)», so, what do they use in literature, and why?

2. PUNCTUAL COMMENTS. Which was the power used for each target?

which is the roughness of the substrate? which is the standard deviation associated to the roughness measurement of the coatings? how many samples were measured for each condition? Authors state that «[roughness] could be an indication of the coatings following the 164 irregularities of the Si substrate during the deposition process (RMS Si = 4nm)...», but they say they used stainless steel (which grade?) as a substrate. Please, use data from only a substrate.

If AFM is not suitable for A1, the authors should use profilometry, to get an idea of all the condition roughness and to compare them (which is not possible by AFM).

You should put some characterization of the substrates (stainless steel, which grade?): surface morphology, roughness, chemical composition. Which is the size of the samples, how they were prepared before deposition?

The scalebars size of figure 2 should be increased, because they are barely visible.
Then, figure 2(a) and 2(f), why do you have this morphology? some more analyses, with a different magnification, are needed, to show another perspective on this specific morphology. And for figure 2(f), why did you get those cavities?

In table 1, you show the thickness of the deposited layers: how this was obtained? do you have cross-section images? You need them to have an idea about the homogeneity of your coating, especially for some conditions that show a very irregular surface. Figure 2(b) shows some kind of directionality in the coating, why?

«The detected traces of Fe could be residue from 199 the deposition chamber.» very sorry, but this sounds not good. Which is the substrate you use, stainless steel? Which is the voltage you use for the analyses, can Fe be a signal coming from the substrate? Which is the standard deviation associated to measure? Also, light elements such as O should not be considered from a quantitative point of view.

Please, present also A3 and B3 x-ray diffractograms. Please, index the present peaks, as well.

For scratch test images, you need a scalebar, and also you need to define the point (in terms of lenght or load position) where the pictures were taken.

Figure 8(e), conditions A3 and B2 are missing. Please, complete.

Author Response

Dear reviewer,

We are grateful for your careful consideration of this paper, and we very much appreciate your detailed suggestions, which have been very helpful in improving the manuscript. We have revised the paper based on your comments and observations. Several improvements in writing as well as in the correction in grammar, spelling and punctuation have been made. We inserted the rationale of choosing the substrates and discussed several points to the framework of the presented work. Other important results were exemplified, as suggested, by applying profilometry on the SS substrate and A1 sample and by showing cross-sectional SEM images based on which the thickness validation was held. Other information such as the grade substrate, preparation of substrates prior deposition and corrections no figures were done accordingly.

We hope that these changes will meet your requirements (see the attached file).

Sincerely,

The authors

Reviewer 4 Report

JOURNAL: nanomaterials

Manuscript number:

Title: Surface, structural, and mechanical properties enhancement of Cr2O3 and SiO2 co-deposited coatings with W or Be

The title clearly describes the article and the abstract reflects the content of the article.

This paper describes the influence of the addition of W or Be in the elaboration of Si02 or Cr2O3 deposits in a conventional magnetron sputtering. Their morphology, chemical composition and structure are described. The mechanical properties (hardness, stiffness, friction coefficient) of the deposits are explained according to their nature.

1. Introduction

The introduction clearly states the problem being investigated and the objectives of this research.

2. experimental details

The authors should specify or even justify the use of several substrates according to the analyses or tests carried out. The nature of the deposits as well as the properties may change according to the nature of the substrates.

The authors should explain the polishing methods used for the substrates and more precisely the one used for the 304L. Is it only a mechanical polishing or a chemical-mechanical polishing? This may condition the adhesion of the deposits.

It would also be interesting to know the roughness of the substrates before deposition.

It would also be interesting to know the roughness of the substrates before deposition.

The thickness of the B1 deposit is very different from those of the other deposits. However, this can condition all the mechanical property measurements (residual stresses, hardness, rigidity, adhesion, etc.). How then can we compare its mechanical properties with those of other deposits?

3. Results

Line 160 : “The results for the A1 (Cr2O3) sample are not shown herein, due to the high surface roughness that would contribute to high displacement on Z of the cantilever integrated in the measuring setup”.

The maximum roughness of the A1 deposit is therefore greater than 7 microns or 15 microns ? (the maximum extent of the TT2 Quantum AFM is of this order of magnitude depending on the size of the scanner). An estimation by confocal microscopy, for example, would be interesting.

It would also be interesting to know the maximum total roughness parameter to compare the different deposits.

Was a single area of 5 microns x 5 microns scanned or several areas per sample? Could we have a standard deviation on the measurements given in figure 1a?

Line 168: “In comparison with the pure oxides, the presence of Be in the co-deposition configuration (samples A3 and B3) seems to lead to an increase in surface roughness”

How can this be said knowing that the roughness of the Cr2O3 deposit is not given? 

Line 186: “Furthermore, the porosity volume fraction for the B3 configuration was determined 186

in the range of 2-3% by image analysis.”

Could we have the size of the scanned area?

Figure 3: It would be interesting to keep the same X-axis scale for each family of deposits (Cr2O3 or SiO2).

Hardness

It is necessary to specify the metallurgical state of the 304L substrate to interpret its hardness value.

It would be interesting to have the loading-unloading curves of the nanoindentation tests in order to :

- Determine the maximum depth achieved for each indentation. The B1 deposit has a very small thickness. Are the mechanical properties obtained by nanoindentation influenced by the substrate?

- Observe whether cracking modes appear during the tests (deposit toughness).

Young modulus

What is the value of the Poisson's ratio chosen to calculate the stiffness E ?

Figure 8: Scales are difficult to read.

The width of the scratches is different between the deposits. Can this be related to the different mechanical properties of the deposits (H/E) and their thickness (sample B1)?

Sample B1 should be treated separately from the others because it does not have the same thickness. The thickness influences the critical loads.

Line 325:” The bucking failure mode observed for the A1 sample evolved as a response to the mechanical stress generated by the stylus translation, where the evolved localized elongated cracks perpendicular to the scratch pattern and expose the substrate were directly associated with already present interface defects.”

What defects are present at the interface?

Figure 9: Keep the same colour for the same sample number.

The authors should describe the morphology of the samples after tribological tests in order to understand the evolution of the friction coefficient (B2 and A1).

Is the evolution of the coefficient of friction (level and range) due to an evolution of the morphology or the nature of the surface of the samples (presence of a 3rd body)?

Line 340:” An agreement between the H/E ratio and the lower friction coefficient can be observed, especially in the case of samples A3 and B1”

It is not possible to compare with sample B1 as it does not have the same thickness.

Author Response

Dear reviewer,

Thank you very much for the feedback and helpful comments as well as for your time. Also, we would like to thank you for your consideration and we hope that our implemented changes will meet your requirements (please see the attachment).

Sincerely,

The authors

Round 2

Reviewer 3 Report

The paper can be published

Author Response

Dear reviewer,

We are thankful that the changes done on the manuscript based on your detailed suggestions satisfied your requirements and we are aware that those observations were very helpful in improving our manuscript.

Sincerely,

The authors

Reviewer 4 Report

Thanks to the authors for considering some of the remarks. It seems to me that the text could be further improved based on the authors' answers concerning the scratch tests and tribology tests.

The authors should explain the polishing methods used for the substrates and more precisely the one used for the 304L. Is it only a mechanical polishing or a chemical-mechanical polishing? This may condition the adhesion of the deposits.

A: Thank you for your observation. The carbon wafers were polished on diamond-paste to a mirror-like finish. The 304L SS substrates were used by default as purchased from the manufacturer. 

The deposits were made on the stainless steel without any preparation ?

It would also be interesting to know the roughness of the substrates before deposition.

A: Profilometry measurements were carried on the SS substrate prior deposition with the determined values: Ra=0.18  μm, Rz=1.09 μm and Rq= 0.22  μm.  Why not state this in the text?

The thickness of the B1 deposit is very different from those of the other deposits. However, this can condition all the mechanical property measurements (residual stresses, hardness, rigidity, adhesion, etc.). How then can we compare its mechanical properties with those of other deposits?

A: The instrumented indentation measurements performed on the coatings were conducted with low loads (penetration depth below 10% of the coating thickness), in order to minimize or exclude entirely the influence of the substrate on the results concerning the mechanical behavior (at least indentation hardness and elastic modulus).

Why not state this in the text?

Was a single area of 5 microns × 5 microns scanned or several areas per sample? Could we have a standard deviation on the measurements given in figure 1a?

A:  Randomly selected single area of 5µm x 5 µm was scanned for each configuration. Standard deviation was determined based on statistical toolbox included in the dedicated software and added in the RMS plot.

Is this area size representative?

Sample B1 should be treated separately from the others because it does not have the same thickness. The thickness influences the critical loads.

A: We thank the reviewer for the observation. We agree that the thickness has an influence on the critical loads and we will take this into account into future measurements.

the results of the scratch tests for this depot (B1) could be deleted.

The authors should describe the morphology of the samples after tribological tests in order to understand the evolution of the friction coefficient (B2 and A1).

A: Initial friction coefficients with Si3N4 balls sliding against deposited films (B2 and A1) were lower. However, friction coefficients increased as debris formed and transferred to the wear track. The morphology of the wear track changed during the friction test from a smooth to a rough one due to the formed debris

Why not state this in the text?

Is the evolution of the coefficient of friction (level and range) due to an evolution of the morphology or the nature of the surface of the samples (presence of a 3rd body)?

A: We consider that the evolution of the coefficient of friction is influenced mainly by the nature of the sample surface, the presence of the 3rd body as Be being very important.

Why not state this in the text?

Author Response

We would like to thank the reviewer for the detailed remarks and comments that we’ve considered been helpful to improve the quality of the manuscript. In the attached document, we present our response to the reviewer’s observations.
